# The effect of plyometric training on physical performance in youth soccer players: A randomized controlled trial with maturation status as a covariate

Roman Holík[1], Jakub Krejčí[2]*, Mark De Ste Croix[3], Michal Lehnert[1]

1 Department of Sport, Faculty of Physical Culture, Palacký University Olomouc, Olomouc, Czech Republic, 2 Department of Natural Sciences in Kinanthropology, Faculty of Physical Culture, Palacký University Olomouc, Olomouc, Czech Republic, 3 School of Education, Health and Science, University of Gloucestershire, Gloucester, United Kingdom

* jakub.krejci@upol.cz

## Abstract

### Purpose

Plyometric jump training (PJT) is an effective means of developing speed, strength, and neuromuscular parameters in youth athletes. However, the effect of PJT on performance outcomes as a function of biological maturation remains unclear.

### Methods

Employing a 12-week training intervention, 37 youth male soccer players aged 10–18 years were stratified by biological maturity using the Mirwald equation (pre-PHV versus post-PHV) and subsequently randomized to a PJT group (pre-PHV: n = 10, post-PHV: n = 10) or a control group (pre-PHV: n = 7, post-PHV: n = 10). The PJT group trained twice per week, with sessions based on multidirectional and short ground contact time exercises. The following performance outcomes were assessed: 20-m sprint, L-run, broad jump, unilateral triple jump, countermovement jump, reactive strength index, and relative leg stiffness. A three-way ANOVA for repeated measures was used for statistical analysis.

### Results

No significant time × intervention × maturity interaction was detected for any variable (all $p \geq 0.21$). The 20-m sprint improved significantly in both PJT groups ($p \leq 0.002$), whereas the control group improved only in pre-PHV ($p = 0.038$). Performance in the L-run improved across all groups ($p \leq 0.003$), with no specific intervention effect. Broad jump improved only following PJT ($p = 0.001$). Unilateral triple jump showed significant gains in both PJT groups ($p \leq 0.032$). Relative leg stiffness increased in

**Data availability statement:** The dataset underlying this article is available in the Zenodo repository at https://doi.org/10.5281/zenodo.17881834.

**Funding:** This study was supported by the Internal Grant Agency of Palacký University Olomouc, grant number IGA_FTK_2024_013, entitled "Effects of a plyometric training program on young athletes of different biological ages". The funders had no role in study design, data collection and analysis, decision to publish, or preparation of the manuscript. There was no additional external funding received for this study.

**Competing interests:** Roman Holík is a doctoral student at Faculty of Physical Culture, Palacký University Olomouc and a coach for the soccer club FK Stipa, Zlín, Czech Republic, where this research was conducted. The other authors have declared that no competing interests exist. This does not alter our adherence to PLOS ONE policies on sharing data and materials.

both PJT groups ($p \leq 0.003$), while it decreased significantly in the post-PHV control group ($p = 0.013$) and remained unchanged in the pre-PHV control group ($p = 0.68$).

## Conclusions

PJT appears to be a suitable and effective method for developing speed, explosive power and neuromuscular performance in youth male soccer players both before and after PHV. However, our results indicate that the role of PJT in influencing physical qualities during adolescence may be different.

## Introduction

Plyometric jump training (PJT) is widely recognized as an effective method for improving physical performance and enhancing neuromuscular parameters [1]. It is particularly popular among athletes engaged in disciplines that require explosive movements, speed or change of direction (COD) [2]. Its effectiveness is primarily based on the utilization of the stretch-shortening cycle (SSC), which consists of a rapid eccentric phase, a brief amortization phase, and a subsequent concentric contraction [1]. This mechanism enables the efficient use of elastic potential energy and activation of the stretch reflex, thereby maximizing explosive force production within a minimal time frame [2]. While the performance of the SSC is partially influenced by genetic factors, it is also highly adaptable to training interventions [3,4]. In SSC-based training such as PJT, neuromuscular parameters play a pivotal role, as they are closely linked not only to athletic performance but also to injury prevention [5,6].

Importantly, when appropriately supervised and progressively dosed, PJT is regarded as an effective and safe training modality for youth athletes [7]. Current knowledge regarding the effects of PJT in youth is largely based on chronological age [8]. Nevertheless, accumulating evidence indicates that biological maturation conceptualized as an internal moderator variable plays a crucial role in shaping training-induced adaptations in youth athletes. Recent systematic reviews have highlighted a lack of studies examining the effects of PJT with respect to biological maturation status [2,9,10]. To provide more robust evidence, further data are required with a focus on biological maturation, particularly estimated peak height velocity (PHV), a key marker for designing training interventions given the profound hormonal and morphological changes occurring during this period [10]. Yet, the timing and tempo of PHV may substantially differ from chronological age, reflecting considerable inter-individual variability in biological maturity [11]. Especially in the post-PHV period, there is still a lack of relevant data that could guide the optimization of plyometric training programs [10].

Additional methodological limitations in the current literature on PJT effects in youth include insufficient randomization [12], short intervention durations [10,13], and moderate to high risk of bias, all of which reduce the overall strength of evidence [8]. Furthermore, studies frequently fail to report detailed intervention parameters, such as ground contact time (GCT). The type of SSC engagement (fast versus slow) is expected to elicit distinct physiological responses and, consequently, divergent training adaptations [10].

Performance outcomes resulting from PJT have most often been assessed through key tests of explosive capability, such as the countermovement jump (CMJ), squat jump, broad jump (BJ), and drop jump (DJ). Among these, vertical jump tests, particularly CMJ, are the most frequently employed measures [14]. However, findings from meta-analyses remain inconsistent. Some studies report more pronounced improvements in post-PHV athletes [15], whereas others have not found significant differences between pre- and post-PHV groups [10]. In horizontal force production, PJT appears to be more effective in post-PHV individuals, potentially due to the greater demands for absolute force generation and the altered vector orientation of force application [10].

In addition to vertical jump performance, linear sprint speed represents another critical component of explosive capability, particularly relevant to sports such as football, where short-distance acceleration and sprinting ability frequently determine match performance [16,17]. Meta-analytical evidence suggests that PJT can significantly improve sprint performance over distances ranging from 5 to 40 m in youth soccer players [18]. These adaptations are more pronounced when PJT includes exercises emphasizing horizontal force application, as they better replicate the direction of force production required during sprint acceleration [19]. Similarly, for linear sprint performance, no consistent differences have been observed between male maturation groups, which may be explained by the high heterogeneity of samples included in meta-analyses [15].

While linear sprinting ability is a key component of explosive performance, COD represents another decisive factor in multidirectional sports such as football, where frequent accelerations, decelerations, and directional transitions are essential for success [20]. Evidence indicates that PJT can significantly enhance COD performance by reducing GCT, improving muscle power output, and increasing overall movement efficiency [21]. Furthermore, PJT programs incorporating a combination of vertical, horizontal, unilateral, and bilateral jump exercises are recommended to maximize the transfer of training effects to COD performance [22]. The evidence regarding effect PJT on COD performance across maturity is also contradictory. More recent studies favour pre-PHV groups [10], whereas earlier meta-analyses reported greater improvements among older male-adolescents [20].

Beyond traditional performance outcomes, neuromuscular parameters such as reactive strength index (RSI) and leg stiffness provide essential insight into the underlying mechanisms of adaptation to PJT. These metrics reflect the athlete's ability to efficiently utilize the SSC and the functional properties of the muscle–tendon complex. RSI quantifies the capacity to rapidly and effectively exploit the SSC [23,24], whereas leg stiffness represents a critical mechanical characteristic influencing the storage and reutilization of elastic energy [13]. Enhancements in these parameters following PJT indicate improved neuromuscular efficiency, contributing directly to superior explosive jump performance [25]. Although meta-analyses in boys consistently report significant improvements in RSI in both pre- and post-PHV groups compared with controls, no clear differences between groups have been established [21,24]. Importantly, assessments of leg stiffness following PJT remain scarce in adolescent populations, as most available data derive from studies involving adults [13].

Overall, current evidence suggests that the effects of PJT in youth may be moderated by biological maturation, however, findings of previous studies remain inconsistent and do not allow for clear practical recommendations. Thus, findings of other well-designed intervention studies may be beneficial as they will allow PJT to be applied with respect to growth and maturation and optimize the long-term performance development of athletes. Therefore, the aim of the present study is to investigate the relationship between PJT and biological maturation and to determine whether integrating PJT into specific maturation stages can elicit greater adaptive responses in youth soccer players than those driven solely by age related growth and development.

## Methods

### Participants

Forty-five male adolescent soccer players, aged 10–18 years, were recruited from football club classified as trained/developmental according to the categories described by McKay et al. [26]. Inclusion criteria included at least two years of organized soccer training, regular attendance (minimum of three sessions per week), and no history of lower-limb injuries

or other health issues within six months that could impair testing performance. Exclusion criteria included any health limitations preventing safe participation in high-intensity exercise during the intervention, or failure to achieve at least 75% attendance in training sessions throughout the intervention. The experimental part of the study was conducted at Faculty of Physical Culture, Palacký University Olomouc from May 13th, 2024 to August 16th, 2024. Ethical approval was obtained from the Ethics Committee of the Faculty of Physical Culture, Palacký University Olomouc (reference number 10/2024), and informed consent was received from participants and their legal guardians. Before the study begins, both the players and their guardians were also informed about the study procedures, study goals, benefits and potential risks.

## Study design

This study was a 12-week randomized controlled trial designed to examine the impact of biological age on the effectiveness of PJT in adolescent soccer players. Participants were categorized into pre-PHV (pre-peak height velocity) and post-PHV (post-peak height velocity) groups based on maturity offset, estimated using the Mirwald method. Within each group, participants were randomized into PJT group or control group, which maintained its usual soccer-specific training program without PJT. Both groups were coached by the same coaches within their normal training process. Pre- and post-intervention assessments measured performance across linear speed, COD speed, explosive power, RSI, and leg stiffness. An overview of the trial timeline, group allocation, and testing time points is presented in Fig 1. The study protocol was registered at ClinicalTrials.gov with the identification number NCT06406764.

## Testing protocols

Participants underwent a series of standardized pre- and post-intervention tests to assess anthropometric characteristics and athletic performance. All testing was conducted at the same time of day (9:30–12:00) to minimize circadian effects. Players were instructed to sleep at least eight hours prior to testing and to refrain from intensive physical activity for 48 hours beforehand. A familiarization session was conducted one week prior to baseline testing. All tests were performed indoors. All sessions were preceded by a 15-minute habitual warm-up incorporating activation, non-specific drills, and specific preparatory exercises. All testing procedures incorporated a minimum 3-minute rest between tests.

## Anthropometric and maturation assessment

Anthropometric characteristics of participants, including body mass, muscle mass, and body fat percentage were measured using a scale with an integrated stadiometer (WB-380, Tanita, Tokyo, Japan) and bioimpedance device (InBody 770, InBody, Seoul, South Korea). Biological age estimation was conducted using the Mirwald method, which calculates the time distance to PHV based on age, standing height, sitting height, and leg length measurements. The maturity offset was calculated using the following formula for boys [27]: maturity offset $= -9.236 + 0.0002708 \times$ (leg length $\times$ sitting height) $- 0.001663 \times$ (age $\times$ leg length) $+ 0.007216 \times$ (age $\times$ sitting height) $+ 0.02292 \times$ (body mass/standing height $\times 100$).

## Sprint and COD testing

*20 m sprint:* Timing gates (Brower Timing System, Draper, UT, USA) were positioned at 0.5 m and 20.5 m along a straight 20 m course. Players started in a crouched "half-start" position with one foot forward, and on a visual signal, sprinted maximally over the 20 m distance [28]. Each player completed two trials with a rest interval of at least two minutes between attempts, and the average time across both trials was used for analysis. This test, widely used for assessing youth speed, has demonstrated high reliability (ICC: 0.66–0.86) [29].

*L-run (COD):* For the L-run test, timing gates (Brower Timing System, Draper, UT, USA) were used, positioned at the start and end of the course to capture the time taken for players to complete the 'L'-shaped route marked by cones 5 m

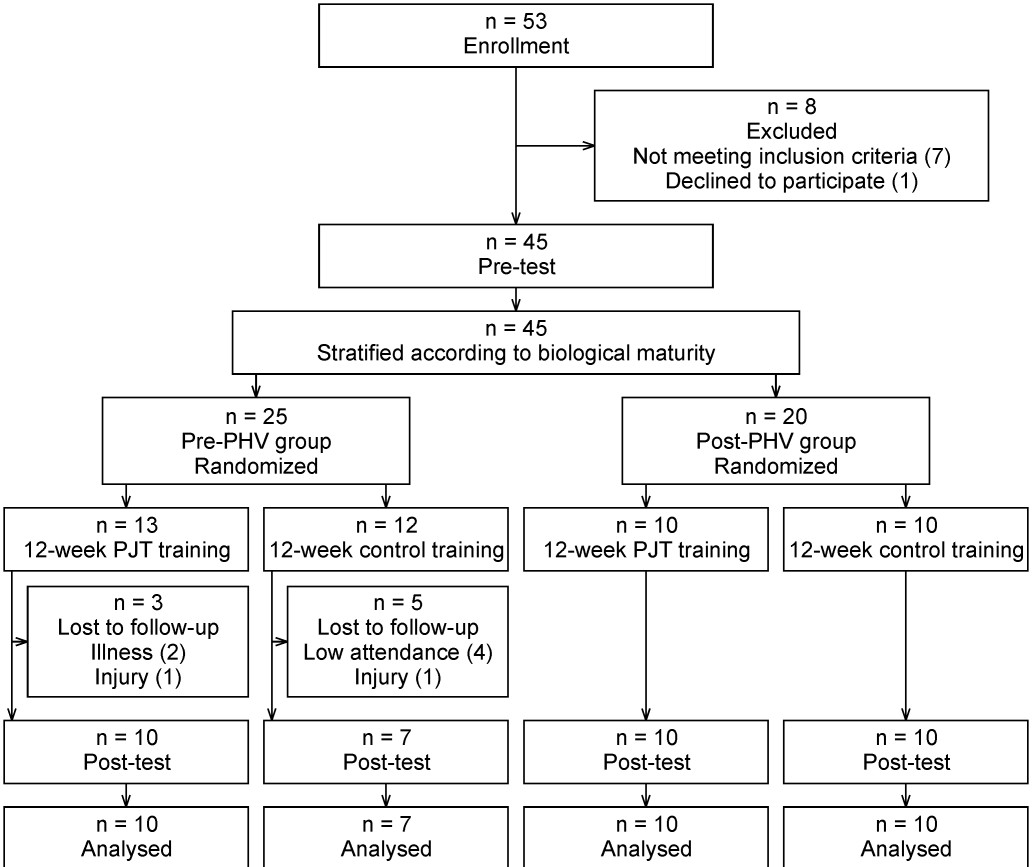

**Fig 1. CONSORT flow diagram.** PHV = peak height velocity; PJT = plyometric jump training.

apart. Players initiated movement from a half-start position, performing the course as quickly as possible while following the directional pattern [30]. Each player completed two attempts, with a rest period of at least one minute, and the average time across both attempts was used for analysis. The L-run has demonstrated high reliability (ICC = 0.94) [31].

## Jumps performance testing

*Countermovement jump:* Players stood with feet shoulder-width apart and hands placed on their hips to control for upper-body involvement. On a verbal signal, they performed a maximal vertical jump with a countermovement, ensuring no arm swing was used to boost jump height [32]. Vertical ground reaction force was measured on two parallel force platforms (AMTI OR6-7-1000, Advanced Mechanical Technology, Watertown, MA, USA) with a sampling frequency of 1000 Hz. A quiet standing period of 2 s was recorded prior to the initiation of each CMJ to ensure an initial velocity of zero. The jump height was calculated from the force-time curve using the impulse-momentum method published by Linthorne [33]. Each player completed three attempts, separated by a one-minute rest. The average value across the three attempts was used for analysis. CMJ shows high reliability (ICC = 0.88) [34].

*Broad jump:* To assess horizontal explosive power, players performed a standing BJ on a flat, marked surface. They started with feet shoulder-width apart behind a designated line, bending at the knees and hips for a countermovement, then leaped forward as far as possible [35]. The jump distance was measured from the start line to the nearest point of heel contact. Each player completed two attempts with a one-minute rest interval and the average distance across both

attempts was used for analysis. This test has high reliability and validity (ICC = 0.96), making it suitable for assessing horizontal explosive power [36].

*Unilateral triple jump:* In the unilateral triple jump (UTJ), participants performed three consecutive maximal hops on one leg along a marked track with a measuring tape. Starting on the dominant leg, they made three forward hops with an emphasis on stability and ended with a two-footed landing. The total distance was measured from the start line to the final heel touchdown. Participants were required to stabilize for two seconds upon landing [37]. Each player completed two attempts per leg. Attempts alternated between legs (right–left–right–left) to minimize unilateral fatigue. The average distance per limb across attempts was used for analysis. Participants rested for a minimum of one minute between attempts. This test has demonstrated high reliability (ICC = 0.97) and is relevant for evaluating single-leg power and stability [38].

### Neuromuscular parameters testing

*Drop jump:* RSI was evaluated using a DJ test from a 30-cm box on force plate (AMTI OR6-7-1000, Advanced Mechanical Technology, Watertown, MA, USA). Participants performed a DJ with hands on hips, stepping off with one foot. Upon landing, they were instructed to a perform a maximal vertical jump as quickly as possible [39]. The GCT and flight time were determined from the force-time curve based on a threshold value of 10 N. The jump height was calculated using a flight time method according to the formula [6]: height = 9.81 × (flight time$^2$)/8. RSI was calculated as the ratio of the jump height to the GCT, in accordance with the procedure described by Flanagan and Comyns [6]. Each participant completed three trials and the average RSI across the three trials was used for analysis. Between repeated jump attempts, a minimum rest interval of one minute was enforced. The RSI test is highly reliable (ICC = 0.96) [40].

*20 submaximal hops:* Using the Optojump system (Optojump Next, Microgate, Bolzano, Italy), players performed 20 repetitive submaximal hops at a frequency of 2.5 Hz, guided by an audio metronome (Wittner, Isny im Allgäu, Germany) to standardize timing. Hands remained on hips to avoid upper body involvement [32]. The middle 10 hops were analyzed for stiffness. Each participant completed three trials and a minimum rest interval of one minute was provided between consecutive trials to minimize fatigue-related effects. Absolute stiffness was calculated using the method by Dalleau et al. [41,42]. Relative leg stiffness (RLS) was then calculated by normalizing the absolute leg stiffness with respect to body mass and leg length, based on the spring-mass model described by McMahon and Cheng [43]. Reliability for this test is high (ICC = 0.93) [32].

### Plyometric training intervention

The plyometric training program was structured based on contemporary research in youth plyometric training, emphasizing performance enhancement and injury prevention [7,10,19,24]. The intervention was carried out during the competitive season and was implemented twice weekly within standard training sessions for 12 weeks. The program was performed after a dynamic warm-up as the main training component. All intervention sessions were led by a single professional coach. Total plyometric work per session was approximately 20 minutes, with at least 48 hours of recovery between sessions. All PJT sessions were performed outdoors on natural grass. Progressive increases in training volume and intensity were prescribed according to Cronin and Radnor [44]. Key principles included: 1. Progressive SSC exposure – progressing from slow SSC (>250 ms) to fast SSC (<250 ms) to develop reactive strength; 2. Exercise variation – bilateral and unilateral movements in vertical, horizontal, and lateral directions. Rest intervals were 60 s during phases 1 and 2, and 90 s during phases 3 and 4. Players were verbally encouraged to minimize GCT during jumps. A detailed phase-by-phase overview, including objectives, is presented in Tables 1 and 2.

### Statistical analysis

Repeated attempts, when available, were averaged for each participant prior to subsequent statistical analysis. The arithmetic mean and standard deviation were used for descriptive purposes. The effect of biological maturity and type of

**Table 1. Plyometric program: phase 1 and phase 2.**

| | Phase 1 | | | | | | Phase 2 | | | | | |
|---|---|---|---|---|---|---|---|---|---|---|---|---|
| Aim | Exercises with extended ground contact time for joint stabilization | | | | | | Transition to shorter ground contact time exercises, emphasizing ankle propulsion | | | | | |
| Exercise | 1. week | | 2. week | | 3. week | | 4. week | | 5. week | | 6. week | |
| | S1 | S2 | S3 | S4 | S5 | S6 | S7 | S8 | S9 | S10 | S11 | S12 |
| Forwards pogos | 2x15 | | 3x15 | | | | | | | | | |
| Backwards pogos | 2x15 | | 3x15 | | | | | | | | | |
| Lateral pogos | 2x15 | | 2x15 | | | | | | | | | |
| Standing hopping | 4x10 | | 3x10 | | 2x10 | | | | | | | |
| Standing hopping + sticks | | 4x10 | | 4x10 | | 3x12 | | | | | | |
| Counter movement jump + stick | | 2x6 | | 3x6 | | 3x8 | | 2x6 | | 2x6 | | 2x6 |
| Fast skipping (2x) + stick | | 2x8 | | 2x8 | | | | | | | | |
| Unilateral forward pogos | | | | | 2x10 | | | | | | | |
| Unilateral backward pogos | | | | | 2x10 | | | | | | | |
| Unilateral lateral pogos | | | | | 2x10 | | | | | | | |
| Fast skipping (3x) + stick | | | | | | 2x8 | 2x8 | | 2x8 | | 2x8 | |
| Broad jump + stick | | | | | | | | 3x5 | | 3x5 | | 3x8 |
| Forwards mini hurdles hops | | | | | | | | 3x5 | | 3x8 | | |
| Lateral mini hurdles hops | | | | | | | | 2x5 | | 2x8 | | |
| Lateral bound to horizontal hops | | | | | | | 3x2 | | 4x2 | | 4x3 | |
| Consequtive horizontal jumps | | | | | | | 2x10 | | 3x10 | | 4x10 | |
| Forwards mini hurdles unilateral hops | | | | | | | | | | | | 2x10 |
| Lateral mini hurdles unilateral hops | | | | | | | | | | | | 2x10 |

S = Sessions.

training on dependent variables was evaluated using analysis of variance (ANOVA) for repeated measures. The 2×2×2 ANOVA included a within-subject factor Time (pre- and post-test), a between-subject factor Intervention (PJT group and control group), and a between-subject factor Maturity (pre- and post-PHV). The full set of interactions was also included. The normality of the residuals was verified using the Shapiro-Wilk test. Sphericity was verified using Mauchly's test. If this assumption was rejected, Greenhouse-Geisser adjustment was used. Pairwise comparison between pre- and post-test were performed using Fisher's least significant difference test if Time factor or interaction with Time was significant. Cohen's d was used to calculate the effect size. The standard deviation in the denominator was calculated as the pooled value of the four standard deviations for each group. The following thresholds recommended for athletes were used [45]: trivial 0.0–0.24, small 0.25–0.49, moderate 0.5–1.0, and large ≥1.0. MATLAB R2024b (MathWorks, Natick, MA, USA) with Statistics Toolbox was used for analyses. The significance level was set at 0.05.

For logistical reasons, the sample size was limited by the availability of participants from a single soccer club. Therefore, no a priori power analysis was conducted. Instead, a posteriori sensitivity analysis was performed using the G*Power software version 3.1.9.7 (Heinrich-Heine-Universität, Düsseldorf, Germany). A two-tailed paired t-test was considered for the calculation, as the Fisher's least significant difference test used in this study for pairwise comparisons is a special type of t-test. The following values were used for the calculation: significance level $\alpha = 0.05$, power $1 - \beta = 0.80$, sample size $n = 10$ or 7. The required effect size was calculated to be $d = 1.0$ or 1.3, respectively.

Table 2. Plyometric program: phase 3 and phase 4.

| | Phase 3 | | | | | | Phase 4 | | | | | |
|---|---|---|---|---|---|---|---|---|---|---|---|---|
| Aim: | Controlled-height or distance jumps with increased lower-limb flexion at the hip and knee joints. | | | | | | High-intensity, maximal-effort jumps with minimal ground contact time, incorporating unilateral high-intensity movements. | | | | | |
| Exercise | 7. week | | 8. week | | 9. week | | 10. week | | 11. week | | 12. week | |
| | S13 | S14 | S15 | S16 | S17 | S18 | S19 | S20 | S21 | S22 | S23 | S24 |
| Unilateral forward pogos | | | | | | | | 2x8 | | 2x8 | | 2x8 |
| Unilateral backward pogos | | | | | | | | | | | | |
| Unilateral lateral pogos | | | | | | | | | | | | |
| Fast skipping (3x) + stick | | | | | | | 2x8 | | 2x8 | | 2x8 | |
| Broad jump + stick | | | | | | | | | | | | |
| Forwards mini hurdles hops | 2x10 | | 2x10 | | | | | | | | | |
| Lateral mini hurdles hops | | 2x10 | | 2x10 | | | | | | | | |
| Lateral bound to horizontal hops | | 4x4 | | 4x5 | | | | | | | | |
| Consequetive horizontal jumps | 4x12 | | 4x12 | | | 4x12 | | | | | | |
| Consequetive horizontal jumps from points to points (1,2 m) | 4x5 | | 4x6 | | | | | | | | | |
| Broad jump | | 4x4 | | 4x5 | | | | | | | | |
| Consequetive horizontal unilateral jumps from points to points (1 m) | | | | | | 2x5 | | | | | | |
| Forwards mini hurdles unilateral hops | | | | | | 2x7 | | | | | | |
| Consequetive broad jumps | | | | | | | 6x3 | | 6x4 | | 6x5 | |
| Unilateral triple jump | | | | | | | | 5x | | 6x | | 7x |

S = Sessions.

## Results

Descriptive characteristics of the study groups are presented in Table 3. Results of the ANOVA evaluating the effects of training are shown in Table 4. Descriptive statistics and pairwise comparison for pre- and post-test are presented in Table 5 and Fig 2. The residuals of all variables followed a normal distribution (all $p \geq 0.11$) except for skeletal muscle mass, for which normality was rejected ($p = 0.036$). After visual inspection of the quantile-quantile plot for this variable, the deviation from normality was assessed as minor, as ANOVA is considered robust for such deviation. Mauchly's test rejected sphericities for all 11 variables studied (all $p < 0.001$), therefore the Greenhouse-Geisser adjustment was used.

### Effect of biological maturity

No significant Time × Intervention × Maturity interaction (all $p \geq 0.21$) and Time × Maturity interaction (all $p \geq 0.16$) was found for any variable (Table 4). Based on the data from our study, we therefore concluded that biological maturity did not significantly moderate the training effect in either PJT group or control group.

### Anthropometric variables

For body mass, a significant ($p < 0.001$) Time factor was found, but no significant ($p = 0.25$) Time × Intervention interaction. Except for an insignificant ($p = 0.41$) change in the post-PHV PJT group, post-hoc analysis found significant increases in body mass in the remaining three groups (all $p \leq 0.035$). However, the effect sizes were trivial (all $d \leq 0.10$) in all four groups.

**Table 3. Characteristics of study groups of adolescent soccer players.**

|  | Pre-PHV | Pre-PHV | Post-PHV | Post-PHV |
|---|---|---|---|---|
|  | PJT | Control | PJT | Control |
| Sample size | 10 | 7 | 10 | 10 |
| Calendar age (years) | 11.7 ± 1.3 | 11.7 ± 0.8 | 16.2 ± 1.5 | 16.0 ± 1.7 |
| Maturity offset (years) | −2.2 ± 0.9 | −1.6 ± 1.0 | 2.0 ± 1.1 | 2.1 ± 1.3 |
| Body height (cm) | 146 ± 8 | 155 ± 11 | 175 ± 4 | 176 ± 8 |
| Sitting height (cm) | 75 ± 3 | 80 ± 4 | 91 ± 3 | 92 ± 4 |

PHV = peak height velocity; PJT = plyometric jump training. The data are presented as arithmetic mean ± standard deviation.

**Table 4. Statistical significance from analysis of variance.**

|  | T x I x M | T x I | T x M | Time | I x M | I | Maturity |
|---|---|---|---|---|---|---|---|
| Body mass | 0.73 | 0.25 | 0.35 | <0.001 | 0.12 | 0.008 | <0.001 |
| Muscle mass | 0.77 | 0.90 | 0.77 | <0.001 | 0.47 | 0.083 | <0.001 |
| Body fat | 0.72 | 0.043 | 0.47 | <0.001 | 0.065 | 0.005 | 0.004 |
| 20-m sprint | 0.86 | 0.041 | 0.16 | <0.001 | 0.022 | 0.079 | <0.001 |
| L-run test | 0.21 | 0.39 | 0.44 | <0.001 | 0.14 | 0.26 | <0.001 |
| Broad jump | 0.74 | 0.060 | 0.94 | <0.001 | 0.18 | 0.20 | <0.001 |
| UTJD | 0.92 | 0.73 | 0.25 | <0.001 | 0.16 | 0.39 | <0.001 |
| UTJN | 0.87 | 0.19 | 0.23 | <0.001 | 0.069 | 0.37 | <0.001 |
| RLS | 0.42 | <0.001 | 0.24 | 0.080 | 0.40 | 0.025 | 0.001 |
| CMJ | 0.93 | 0.68 | 0.89 | 0.024 | 0.24 | 0.11 | <0.001 |
| RSI | 0.51 | 0.13 | 0.16 | 0.47 | 0.13 | 0.19 | <0.001 |

T = time factor; I = intervention factor; M = maturity factor; UTJD = unilateral triple jump on dominant leg; UTJN = unilateral triple jump on non-dominant leg; RLS = relative leg stiffness; CMJ = countermovement jump height; RSI = reactive strength index.

For skeletal muscle mass, a significant ($p < 0.001$) Time factor was found, but no significant ($p = 0.90$) Time × Intervention interaction. Post-hoc analysis revealed significant (all $p \leq 0.023$) increases in muscle mass in all four groups studied, but the effect sizes were trivial (all $d \leq 0.20$).

For body fat, a significant ($p < 0.001$) Time factor and a significant ($p = 0.043$) Time × Intervention interaction was found. Post-hoc analysis showed significant (both $p \leq 0.005$) decreases in body fat only in the PJT groups. In the control groups, the decreases were not significant (both $p \geq 0.20$). However, the effect sizes in all four groups were trivial (all absolute $d \leq 0.23$).

## Sprint and COD performance

For the 20-m sprint, a significant ($p < .001$) Time factor was found with a significant ($p = 0.041$) Time × Intervention interaction. Post-hoc analysis showed that improvements in the 20-m sprint were significant (both $p \leq 0.002$) in the PJT groups with medium effect sizes ($d = −0.77$ and $−0.55$ for pre-PHV and post-PHV, respectively). In the control groups, the improvement was significant with a small effect size ($p = 0.035$, $d = −0.43$) only for pre-PHV subgroup, but the change was not significant with a trivial effect size ($p = 0.38$, $d = −0.15$) for post-PHV group.

**Table 5. The effect of plyometric and control training on anthropometric, speed, power, and neuromuscular variables in groups pre- and post-peak height velocity.**

|  |  | Pre-PHV | Pre-PHV | Post-PHV | Post-PHV |
|---|---|---|---|---|---|
|  |  | PJT | Control | PJT | Control |
| Body mass (kg) | T1 | 38±8 | 54±21 | 64±7 | 68±6 |
|  | T2 | 39±9 | 55±20 | 64±7 | 69±5 |
|  | $p$ ($d$) | 0.035 (0.07) | 0.015 (0.10) | 0.41 (0.03) | 0.020 (0.08) |
| Muscle mass (kg) | T1 | 17±3 | 20±5 | 32±4 | 33±3 |
|  | T2 | 18±3 | 21±5 | 33±4 | 34±4 |
|  | $p$ ($d$) | 0.008 (0.19) | 0.017 (0.20) | 0.008 (0.19) | 0.023 (0.16) |
| Body fat (%) | T1 | 14±6 | 25±14 | 11±3 | 13±4 |
|  | T2 | 12±6 | 24±14 | 10±2 | 13±4 |
|  | $p$ ($d$) | 0.002 (−0.23) | 0.20 (−0.10) | 0.005 (−0.20) | 0.69 (−0.03) |
| 20-m sprint (s) | T1 | 3.95±0.30 | 4.23±0.38 | 3.36±0.13 | 3.27±0.11 |
|  | T2 | 3.77±0.25 | 4.13±0.37 | 3.23±0.15 | 3.23±0.14 |
|  | $p$ ($d$) | <0.001 (−0.77**) | 0.038 (−0.43*) | 0.002 (−0.55**) | 0.38 (−0.15) |
| L-run test (s) | T1 | 6.83±0.57 | 7.17±0.76 | 5.94±0.23 | 5.89±0.23 |
|  | T2 | 6.46±0.59 | 6.94±0.75 | 5.70±0.21 | 5.63±0.23 |
|  | $p$ ($d$) | <0.001 (−0.79**) | 0.003 (−0.50**) | <0.001 (−0.51**) | <0.001 (−0.56**) |
| Broad jump (cm) | T1 | 166±29 | 152±26 | 221±9 | 224±20 |
|  | T2 | 179±21 | 157±28 | 233±11 | 230±20 |
|  | $p$ ($d$) | 0.001 (0.60**) | 0.26 (0.22) | 0.001 (0.56**) | 0.080 (0.29*) |
| UTJD (cm) | T1 | 460±78 | 409±89 | 628±29 | 642±77 |
|  | T2 | 491±77 | 436±82 | 648±46 | 660±76 |
|  | $p$ ($d$) | 0.001 (0.44*) | 0.014 (0.38*) | 0.032 (0.28*) | 0.057 (0.24) |
| UTJN (cm) | T1 | 454±80 | 396±98 | 632±34 | 661±77 |
|  | T2 | 486±77 | 417±83 | 655±45 | 670±56 |
|  | $p$ ($d$) | 0.001 (0.44*) | 0.048 (0.29*) | 0.015 (0.31*) | 0.32 (0.12) |
| RLS (kN/m) | T1 | 25.9±4.0 | 23.8±2.3 | 29.9±5.5 | 30.9±4.8 |
|  | T2 | 29.5±2.4 | 23.3±1.8 | 33.1±5.6 | 28.3±3.9 |
|  | $p$ ($d$) | 0.001 (0.81**) | 0.68 (−0.11) | 0.003 (0.72**) | 0.013 (−0.58**) |
| CMJ (cm) | T1 | 21.6±4.0 | 17.3±4.5 | 31.7±4.3 | 31.2±6.4 |
|  | T2 | 22.5±4.0 | 18.0±5.0 | 32.9±3.8 | 32.0±5.8 |
|  | $p$ ($d$) | 0.19 (0.19) | 0.41 (0.14) | 0.12 (0.22) | 0.30 (0.15) |
| RSI (m/s) | T1 | 0.79±0.25 | 0.59±0.22 | 1.16±0.21 | 1.24±0.29 |
|  | T2 | 0.75±0.24 | 0.50±0.18 | 1.24±0.22 | 1.19±0.35 |
|  | $p$ ($d$) | 0.51 (−0.15) | 0.19 (−0.37*) | 0.13 (0.36*) | 0.43 (−0.19) |

PHV = peak height velocity; PJT = plyometric jump training; T1 = pre-test; T2 = post-test; p = significance of Fisher's least significant difference test; d = Cohen's d; UTJD = unilateral triple jump on dominant leg; UTJN = unilateral triple jump on non-dominant leg; RLS = relative leg stiffness; CMJ = countermovement jump height; RSI = reactive strength index; * = small; ** = medium; *** = large effect size. The data are presented as arithmetic mean ± standard deviation.

For the L-run test, a significant ($p < 0.001$) Time factor and an insignificant ($p = 0.39$) Time × Intervention interaction was found. Post-hoc analysis revealed significant (all $p \leq 0.003$) improvements in the L-run test with medium effect sizes ($d$ ranging from −0.79 to −0.50) in all four groups.

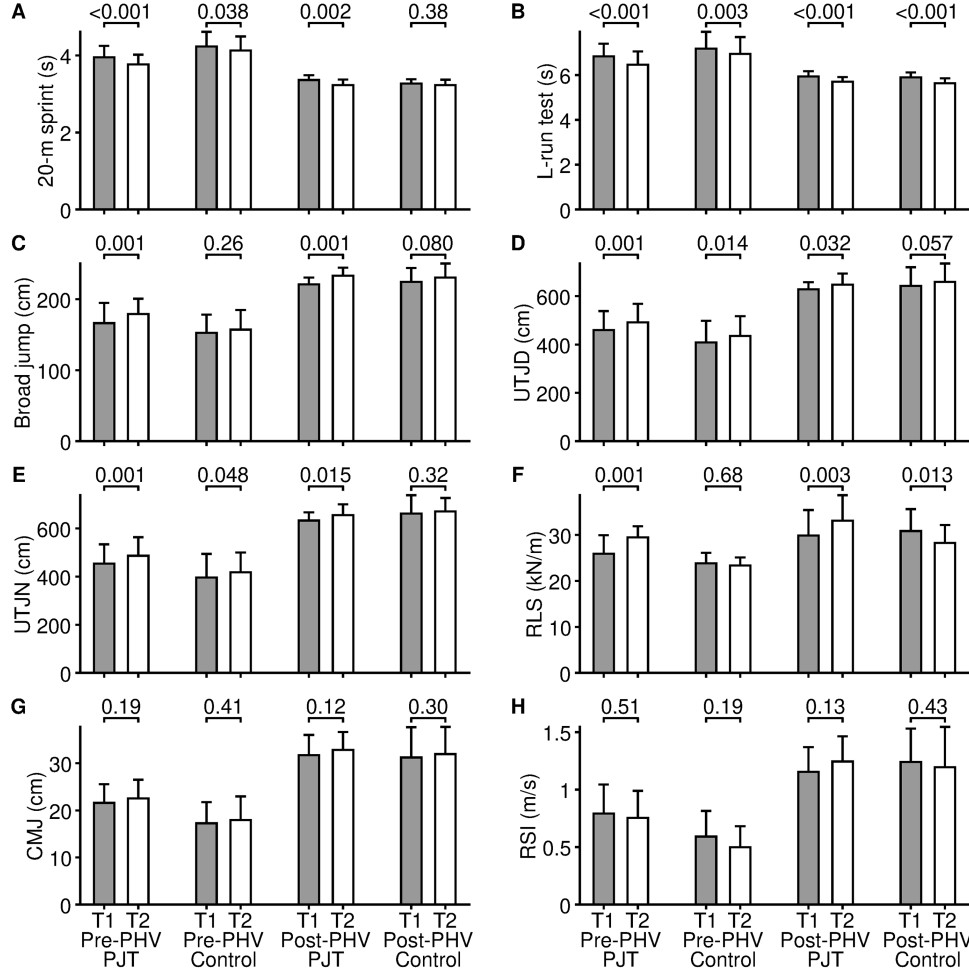

**Fig 2. The effect of plyometric jump training (PJT) and control training on 20-m sprint (A), L-run test (B), broad jump (C), unilateral triple jump dominant (D), unilateral triple jump non-dominant (E), relative leg stiffness (F), countermovement jump height (G), and reactive strength index (H) in groups pre- and post-peak height velocity (PHV).** T1 = pre-test; T2 = post-test. The data are presented as arithmetic mean and standard deviation. The values displayed above the bars are significances of Fisher's least significant difference test.

## Jump performance

For the BJ, a significant ($p < 0.001$) Time factor and an insignificant ($p = 0.060$) Time × Intervention interaction was found. However, the $p$-value of the interaction was close to the significance level, and therefore a post-hoc analysis found significant (both $p = 0.001$) improvements in BJ only in the PJT groups with medium effect sizes ($d = 0.60$ and 0.56). The changes were not significant (both $p \leq 0.080$) in the control groups with a trivial ($d = 0.22$) and small ($d = 0.29$) effect size for pre-PHV and post-PHV, respectively.

For the CMJ height, a significant ($p = 0.024$) Time factor and an insignificant ($p = 0.68$) Time × Intervention interaction were found. Despite the significant Time factor, no significant (all $p \geq 0.12$) change in CMJ height was found in any of the four groups in the post-hoc analysis. In addition, the effect sizes were trivial (all $d \leq 0.22$).

The findings in UTJ dominant and UTJ non-dominant were similar. The Time factor was significant (both $p < 0.001$) and the Time × Intervention interaction was not significant ($p = 0.73$ and 0.19 for the dominant and non-dominant, respectively). In the PJT groups and the pre-PHV control group, improvements in UTJs were significant (all $p \leq 0.048$) with small effect

sizes ($d$ ranging from 0.28 to 0.44), as shown by post-hoc analysis. In the post-PHV control group, changes in UTJs were not significant (both $p \geq 0.057$) with trivial effect sizes (both $d \leq 0.24$).

### Neuromuscular parameters

For the RLS, a unique finding was observed compared to the other variables studied. The Time × Intervention interaction was significant ($p < 0.001$), but the Time factor alone was not significant ($p = 0.080$). Post-hoc analysis showed that both PJT groups significantly (both $p \leq 0.003$) improved RLS with medium effect sizes ($d = 0.81$ and 0.72 for pre- and post-PHV, respectively). In contrast, both control groups worsened RLS, with an insignificant and trivial change ($p = 0.68$, $d = -0.11$) in the pre-PHV group and a significant and medium change ($p = 0.013$, $d = -0.58$) in the post-PHV group.

The RSI was the only variable in this study for which neither the Time factor ($p = 0.47$) nor the Time × Intervention interaction ($p = 0.13$) was significant. Effect sizes were trivial to small in all four groups ($d$ ranging from $-0.37$ to 0.36).

## Discussion

Our findings demonstrate that PJT induces comparable adaptations in sprint, COD, jumps performance and neuromuscular performance in pre- and post-PHV soccer players. However, while statistical analysis did not reveal a significant time × intervention × maturity interaction, a closer inspection of the data suggests that the drivers of these improvements may differ between groups. In the pre-PHV cohort, improvements in several parameters were observed in both the PJT and control groups, suggesting that PJT acted synergistically with natural growth and maturation. In contrast, the post-PHV control group showed stagnation or even decline in certain variables (RLS), implying that PJT was a critical stimulus to counteract these trends and drive performance enhancements. Collectively, these outcomes highlight the robust efficacy of PJT as a stimulus capable of eliciting meaningful neuromuscular enhancements throughout adolescence, reinforcing its relevance as a systematic training component across both stages of biological maturity.

### Sprint and COD performance

For sprint performance, we observed a significant time × intervention interaction, indicating a clear effect of the intervention on improving linear sprint speed, with a medium effect size in both experimental groups. However, distinct patterns were observed in the control groups. The pre-PHV control group significantly improved its 20 m sprint performance, likely benefitting from natural maturation related increases in force and coordination. In contrast, the post-PHV control group showed no significant improvement. This suggests that while pre-PHV players may experience speed gains through natural development alone, post-PHV players rely more heavily on the specific training stimulus provided by PJT to achieve further improvements.

In contrast, no such effect was evident in the L-run, where all groups improved over time with medium effect sizes observed in all groups. While the time × intervention × maturity interaction was not statistically significant, the divergent trends in the sprint control group data imply that biological maturation moderates the necessity of the training stimulus. For older adolescents, PJT appears essential to overcome the plateau in natural speed development.

The improvement in sprint performance after PJT is in line with recent meta-analyses reporting enhanced sprint speed in young athletes across different ages and maturational stages [10]. In contrast, Asadi et al. [20] have suggested greater effects in older adolescents. However, those findings were based on chronological age rather than precise assessments of biological maturity, which may explain the conflicting conclusions. Our data therefore support the notion that PJT exerts universal benefits across maturational periods and reinforces the value of systematically implementing PJT in both pre- and post-PHV players.

Sprint improvements can be primarily explained by neuromuscular adaptations stimulated by PJT [46]. In pre-PHV athletes, high neuromuscular plasticity likely plays a central role, including more efficient motor unit recruitment, improved synchronization of muscle fibers, and enhanced inter and intramuscular coordination during the SSC, thereby facilitating

motor learning [10]. In contrast, in post-PHV players, morphological adaptations appear to contribute more substantially. Increases in muscle architecture, such as greater thickness and fascicle length, enable higher absolute force production and more effective propulsion [47]. Older adolescents are also better able to orient force vectors horizontally, which is a key determinant of acceleration and maximal speed performance [48–50]. This ability may have allowed post-PHV players to utilize the program's content more efficiently, ultimately resulting in comparable training effects across maturational groups.

Regarding COD, our results showed no significant differences in PJT effects between pre- and post-PHV players, which contrasts with the meta-analysis by Ramirez-Campillo, Sortwell et al. [10], who reported larger effects in pre-PHV athletes. Conversely, earlier findings by Asadi et al. [20] indicated greater improvements in older adolescents, although, again, those analyses were based on chronological rather than biological age, potentially accounting for discrepancies. Ramirez-Campillo, Sortwell et al. [10] further highlighted that PJT effects on COD are often nonsignificant, largely due to small sample sizes, heterogeneity of training protocols, and the use of active controls. This aligns with our methodology, where control group continued to engage in regular football training, which naturally develops COD ability and may have masked the additional effect of our intervention.

Another explanation lies in the specificity of our program, which emphasized fast SSC with short GCT and ankle-dominant rebounds but likely lacked sufficient exercises specifically targeting force-dominant COD (>90°). Such tasks require longer GCT and greater ranges of motion at the knee and hip joints, and strong eccentric braking impulses, often associated with the quality of the penultimate step and control of knee valgus [51,52]. Recent work also emphasizes that dividing SSC solely into "fast" and "slow" categories is overly simplistic, as COD performance is determined by an interplay of factors including movement direction, motor pattern (e.g., lateral), range of motion, and the balance between braking and re-acceleration phases [22]. These insights suggest that if the goal is to enhance sharp COD performance, PJT must systematically include drills with longer contact times and greater ranges of motion, rather than relying exclusively on fast, ankle-dominant rebounds.

## Jump performance

In the BJ (with a medium effect size) and unilateral triple jump (with a small effect size), both experimental groups improved significantly over time, whereas the control groups showed no changes. However, a distinct maturational trend emerged in the UTJ. The pre-PHV control group achieved significant improvements in both the dominant and non-dominant legs, likely benefiting from natural improvements in neuromuscular coordination and limb control associated with this developmental stage. However, the post-PHV control group did not show significant improvements in UTJ performance. In contrast, changes in CMJ performance were small and statistically nonsignificant in both PJT groups. The time × intervention × maturity interaction was not significant for any of the assessed variables.

A recent meta-analysis examining the effects of PJT in youth team sports (predominantly soccer) reported significant improvements in CMJ and standing long jump across maturational stages, with the largest gains observed in post-PHV players [21]. Similarly, Ramirez-Campillo et al. [10], based on data from 11 studies, concluded that PJT improves a wide range of jump-related performance indicators regardless of maturity, with small-to-moderate effect sizes. Supporting this, Asadi et al. [20] demonstrated significant improvements in both BJ and vertical jump among young soccer players of all maturational stages after a 6-week PJT program, with the greatest gains in post-PHV athletes. These findings partially align with our results and reinforce the conclusion that PJT effectively develops jump performance independent of maturational status. Importantly, despite systematically searching the literature, we found no previous studies examining multi-step horizontal unilateral jumps in relation to biological maturation. Our results for UTJ dominant and UTJ non-dominant therefore provide the first data in this area, suggesting that such performance can also be effectively improved through PJT in both pre- and post-PHV periods.

The absence of a significant between-group effect of intervention may have been explained by the limited sample size and the use of active control group, a limitation repeatedly highlighted in the literature [10]. From a practical standpoint,

however, this does not indicate a lack of training effect, as both PJT groups improved over time, whereas control groups did not. Thus, our findings support the conclusion that PJT exerts comparable effectiveness across maturational stages, albeit through different mechanisms. It has been previously suggested that pre-PHV athletes, improvements are likely driven by the synergistic effect of natural growth and training stimulus, combined with high neuromuscular plasticity, motor learning, and more efficient utilization of the SSC, even without major morphological adaptations. In contrast, in post-PHV athletes, hormonally driven morphological adaptations, hypertrophy, and a more developed neuromuscular system appear to play a greater role [2].

Test specificity may further explain the differential outcomes between the BJ and CMJ. While the BJ and triple jump reflect the ability to generate force in the horizontal direction, the CMJ primarily evaluates vertical force production. Jiménez-Reyes et al. [53] showed that vertical jumps may not adequately capture neuromuscular adaptations associated with horizontal force application, which are more evident in sprinting performance. This may explain why our study observed more pronounced improvements in horizontally oriented tests compared with CMJ. An additional factor is the high intra-individual variability and lower sensitivity of CMJ height to detect subtle neuromuscular changes, as emphasized in recent work [54].

The importance of training specificity is further supported by Laurent, Baudry, and Duchateau [55], who demonstrated that the type of plyometric exercise critically determines the transfer of adaptations. Fast rebounds with minimal knee flexion primarily increased leg stiffness, whereas exercises with larger ranges of motion and greater knee and hip involvement were more effective in improving CMJ performance. This may explain why our program elicited significant improvements in BJ and triple jump, while changes in CMJ remained limited.

## Neuromuscular parameters

For RSI, we did not observe any meaningful intervention effect, and none of the groups improved over time, nor did we observe any meaningful effect according to Cohen's $d$. In contrast, a main effect was also confirmed for RLS, which significantly increased in both PJT groups, with a medium effect size. However, a striking difference was observed between the control groups. While the pre-PHV control group maintained their RLS, the post-PHV control group exhibited a significant decrease, also with a medium effect size. These findings suggest that RSI was not sufficiently sensitive to our training stimulus, whereas RLS responded clearly to PJT, and the absence of specific plyometric loading may lead to its deterioration in post-PHV players. The time × intervention × maturity interaction was not significant for either RSI or RLS.

A similar pattern was reported by Lloyd et al. [56], who observed significant increases in leg stiffness among 12- and 15-year-old boys (mean maturity offset −1.8 and +1.1 years from PHV) following a 4-week PJT program, whereas stiffness decreased in 15-year-old controls. RSI improved only in the 12-year-old group, but not in the youngest or oldest participants, supporting our observation that RSI does not respond consistently across age or maturational status. In contrast, Uzelac-Sciran et al. [57] found that eight weeks of jump training elicited a large improvement in RSI in pre-PHV boys, while the response in post-PHV players was small and statistically nonsignificant. These discrepancies may be partly explained by differences in study populations: both cited studies were conducted in general physical education settings, whereas our participants were sport-specific trained athletes.

The literature generally assumes that in non-athletic populations, adolescence is associated with a gradual increase in stiffness due to gains in muscle mass and improvements in neuromuscular coordination [56,58]. However, in young athletes, this developmental trajectory may differ. De Ste Croix et al. [59] demonstrated that RLS significantly decreased after PHV in elite youth team-sport players, a trend also observed in our post-PHV control group. Rapid increases in body mass and alterations in body composition during adolescence may transiently impair inter- and intramuscular coordination and, consequently, the ability to absorb and reutilize elastic energy, unless counteracted by targeted plyometric loading. Our results highlight that a well-designed PJT program can reverse this negative trend and enhance RLS post-PHV, which has important implications not only for performance but also for injury prevention.

RSI and RLS represent distinct components of SSC function and may respond differently to training stimuli. As outlined by Lehnert et al. [60], RSI primarily reflects the ability to rapidly react and utilize fast SSC actions with extremely short GCT (e.g., drop jumps), whereas RLS reflects the capacity to repeatedly absorb and return elastic energy during submaximal loads with longer contact times. The specific design of our program, which included primarily submaximal hops and rebounds, may therefore have preferentially stimulated adaptations related to RLS rather than RSI. It is plausible that an alternative training protocol incorporating a greater volume of maximally reactive jumps with minimal GCT would have elicited more pronounced changes in RSI. The divergent responses of RSI and RLS thus reflect their distinct physiological characteristics. Our findings suggest that PJT emphasizing fast but submaximal jumps may be particularly effective for improving RLS, while targeted inclusion of maximal reactive drills may be necessary to achieve meaningful improvements in RSI.

## Limitations

The interpretation of our findings is primarily constrained by the sample size, in particular the number of participants in the pre-PHV control group was low (n = 7). Participants were further categorized into pre- and post-PHV groups based on biological maturation estimated via the Mirwald method, whose maturity offset carries an approximate error of ±0.5 years. Another limitation of the present study is the absence of a circa-PHV group, which limits the ability to assess how players around peak height velocity might have responded compared to pre-PHV and post-PHV groups. Finally, the intervention program was designed identically for players pre- and post-PHV, and it cannot be ruled out that targeted protocols tailored to the specific needs of distinct maturational phases would have yielded different magnitudes of adaptation.

## Conclusions

A 12-week PJT program produced significant improvements in sprint, horizontal jumping, and RLS performance in youth soccer players disregards maturity status. This finding supports suggestion that PJT is an effective tool for developing physical fitness qualities throughout adolescence. While the overall magnitude of adaptation was comparable across maturity stages, observed trends in control groups point to the different role of PJT in development of physical qualities. In pre-PHV players, PJT acted synergistically with natural growth to amplify gains, whereas in the post-PHV cohort, it was essential to counteract the observed stagnation in speed and the significant decline in relative leg stiffness. No meaningful changes were observed in CMJ and RSI, which implies that PJT adaptations are parameter-specific and may not manifest uniformly across all performance tests.

## Practical applications

We recommend integrating PJT across the entire adolescent period, in both pre- and post-PHV players. However, practitioners should understand that while in pre-PHV players PJT acts synergistically with natural growth to amplify performance gains. In contrast, for post-PHV athletes, PJT is not merely an enhancement tool but a critical necessity to overcome the plateau in natural speed development and, crucially, to counteract the decline in RLS that can occur in the absence of targeted loading. When the objective is to enhance specific outcomes such as CMJ height or RSI, coaches should design targeted PJT, aligning exercise selection characteristics with the desired adaptation profile.

## Acknowledgments

The authors would like to express their sincere gratitude to the staff of the Faculty of Physical Culture, Palacký University Olomouc, for their invaluable assistance during the testing and data collection phases. Special thanks are extended to the professional staff at the Application Centre BALUO for providing technical support and access to their facilities.

Furthermore, we are deeply grateful to all the youth soccer players and their legal guardians for their cooperation, commitment, and enthusiastic participation in this 12-week study.

## Author contributions

**Conceptualization:** Roman Holík, Mark De Ste Croix, Michal Lehnert.

**Data curation:** Roman Holík, Jakub Krejčí.

**Formal analysis:** Jakub Krejčí.

**Funding acquisition:** Roman Holík.

**Investigation:** Roman Holík, Jakub Krejčí.

**Methodology:** Roman Holík, Michal Lehnert.

**Project administration:** Roman Holík.

**Resources:** Roman Holík.

**Supervision:** Michal Lehnert.

**Visualization:** Jakub Krejčí.

**Writing – original draft:** Roman Holík, Jakub Krejčí.

**Writing – review & editing:** Mark De Ste Croix, Michal Lehnert.

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
