## [Decision Letter · Decision Letter 0]

8 Apr 2026

PONE-D-26-10103The effect of plyometric training on physical performance in youth soccer players: A randomized controlled trial with maturation status as a covariatePLOS One

Dear Dr. Krejčí,

Thank you for submitting your manuscript to PLOS ONE. After careful consideration, we feel that it has merit but does not fully meet PLOS ONE’s publication criteria as it currently stands. Therefore, we invite you to submit a revised version of the manuscript that addresses the points raised during the review process.

**We received the comments from the two reviewers. Both suggested the paper to be accepted, but some of their comments need to be addressed before fully considering the paper for publication. Please find their comments below.**

We look forward to receiving your revised manuscript.

Kind regards,

Stefano Amatori, Ph.D.

Academic Editor

PLOS One

**Journal Requirements:**

1. When submitting your revision, we need you to address these additional requirements. Please ensure that your manuscript meets PLOS ONE's style requirements, including those for file naming. The PLOS ONE style templates can be found at https://journals.plos.org/plosone/s/file?id=wjVg/PLOSOne_formatting_sample_main_body.pdf and https://journals.plos.org/plosone/s/file?id=ba62/PLOSOne_formatting_sample_title_authors_affiliations.pdf 2. Thank you for stating in your Funding Statement: This study was supported by the Internal Grant Agency of Palacký University Olomouc (URL: www.upol.cz), grant number IGA_FTK_2024_013, entitled “Effects of a plyometric training program on young athletes of different biological ages”. Please provide an amended statement that declares *all* the funding or sources of support (whether external or internal to your organization) received during this study, as detailed online in our guide for authors at http://journals.plos.org/plosone/s/submit-now. Please also include the statement “There was no additional external funding received for this study.” in your updated Funding Statement. Please include your amended Funding Statement within your cover letter. We will change the online submission form on your behalf. 3. Thank you for stating the following financial disclosure: This study was supported by the Internal Grant Agency of Palacký University Olomouc (URL: www.upol.cz), grant number IGA_FTK_2024_013, entitled “Effects of a plyometric training program on young athletes of different biological ages”.   Please state what role the funders took in the study.  If the funders had no role, please state: "The funders had no role in study design, data collection and analysis, decision to publish, or preparation of the manuscript." If this statement is not correct you must amend it as needed. Please include this amended Role of Funder statement in your cover letter; we will change the online submission form on your behalf. 4. Thank you for stating the following in the Competing Interests section:  I have read the journal's policy and the authors of this manuscript have the following competing interests: Roman Holík is a coach at the soccer club (FK Stipa, Zlin, Czech Republic) where the research was conducted. The other authors have declared that no competing interests exist.We note that one or more of the authors are employed by a commercial company.  a. Please provide an amended Funding Statement declaring this commercial affiliation, as well as a statement regarding the Role of Funders in your study. If the funding organization did not play a role in the study design, data collection and analysis, decision to publish, or preparation of the manuscript and only provided financial support in the form of authors' salaries and/or research materials, please review your statements relating to the author contributions, and ensure you have specifically and accurately indicated the role(s) that these authors had in your study. You can update author roles in the Author Contributions section of the online submission form. Please also include the following statement within your amended Funding Statement. “The funder provided support in the form of salaries for authors, but did not have any additional role in the study design, data collection and analysis, decision to publish, or preparation of the manuscript. The specific roles of these authors are articulated in the ‘author contributions’ section.”If your commercial affiliation did play a role in your study, please state and explain this role within your updated Funding Statement.  b. Please also provide an updated Competing Interests Statement declaring this commercial affiliation along with any other relevant declarations relating to employment, consultancy, patents, products in development, or marketed products, etc.   Within your Competing Interests Statement, please confirm that this commercial affiliation does not alter your adherence to all PLOS ONE policies on sharing data and materials by including the following statement: "This does not alter our adherence to  PLOS ONE policies on sharing data and materials.” (as detailed online in our guide for authors http://journals.plos.org/plosone/s/competing-interests) . If this adherence statement is not accurate and  there are restrictions on sharing of data and/or materials, please state these. Please note that we cannot proceed with consideration of your article until this information has been declared. Please include both an updated Funding Statement and Competing Interests Statement in your cover letter. We will change the online submission form on your behalf. 5. Thank you for stating the following in the Competing Interests section: I have read the journal's policy and the authors of this manuscript have the following competing interests: Roman Holík is a coach at the soccer club (FK Stipa, Zlin, Czech Republic) where the research was conducted. The other authors have declared that no competing interests exist.  Please confirm that this does not alter your adherence to all PLOS ONE policies on sharing data and materials, by including the following statement: "This does not alter our adherence to  PLOS ONE policies on sharing data and materials.” (as detailed online in our guide for authors http://journals.plos.org/plosone/s/competing-interests). If there are restrictions on sharing of data and/or materials, please state these. Please note that we cannot proceed with consideration of your article until this information has been declared.  Please include your updated Competing Interests statement in your cover letter; we will change the online submission form on your behalf. 6. If the reviewer comments include a recommendation to cite specific previously published works, please review and evaluate these publications to determine whether they are relevant and should be cited. There is no requirement to cite these works unless the editor has indicated otherwise.

Reviewers' comments:

Reviewer's Responses to Questions

**Comments to the Author**

1. Is the manuscript technically sound, and do the data support the conclusions?

Reviewer #1: Yes

Reviewer #2: Yes

2. Has the statistical analysis been performed appropriately and rigorously?

Reviewer #1: Yes

Reviewer #2: Yes

3. Have the authors made all data underlying the findings in their manuscript fully available?

Reviewer #1: Yes

Reviewer #2: Yes

4. Is the manuscript presented in an intelligible fashion and written in standard English?

Reviewer #1: Yes

Reviewer #2: Yes

5. Review Comments to the Author

**Reviewer #1:** Dear authors,

Thank you for your submission. I think it was well written.

I have a small number of recommendations.

Method

Please include a reference for all test protocols - e.g. a citation for CMJ line 188 and do the same for all tests.

Please add equations and calculations for:

- Boys PHV - Malian et al. 2020

- CMJ height - the Optojump uses the flight-time method outlined by Glatthorn 2011

- RSI mm/ms outlined by Flanagan and Comyns 2008

- Leg stiffness - equation can be found in the leg stiffness in female

Soccer players by De Ste Croix and Lehrert

- Relative leg stiffness - McMahon and Cheng 1990

I think doing this would add to the rigour of the manuscript.

You should think about adding an abbreviation section - refer to Lloyd et al. 2009.

Statistical analysis

How was the sample size determined? G*Power? Detail the exact procedure. Please add this to

Statical analysis section.

Discussion

I think you should include some the data from the r was ultra to back up discussion points.

Also, i think you should cite some of the adaptations following PT - refer to Markovic and Mikulic 2010.

Limitations

A further limitation was that the study didn't include a mid-PHV. Can you detail what this

Might mean - would the results be similar in the mid-PHV group? If yes/no, why? Read the meta-analysis by Ramirez-Campillo and Sortwell 2023.

I hope this helps.

**Reviewer #2:** The study aimed to investigate whether the performance benefits of CMJ training are influenced by biological maturation status. This is a relevant and timely research question, particularly considering the importance of biological maturation in youth athletic development and training responsiveness.

However, although maturation status was included as a covariate, designing and implementing the training program according to the participants’ maturation stage might have allowed for a more precise evaluation of training adaptations and potentially revealed differential responses across maturation groups. Such an approach may also strengthen the practical implications for coaches and practitioners working with youth soccer players.

Overall, the manuscript is clearly written and methodologically sound. No further major comments or additional suggestions.

6. PLOS authors have the option to publish the peer review history of their article (what does this mean?). If published, this will include your full peer review and any attached files.

Reviewer #1: **Yes:** Lee David McGarrigal

Reviewer #2: No

---

## [Author Response · Author response to Decision Letter 1]

21 Apr 2026

Reviewer #1

We would like to thank the reviewer for the time he spent reviewing the manuscript and for the constructive feedback. It has helped us improve the quality and clarity of our manuscript. We carefully revised the manuscript according to their comments and have provide a point-by-point response to each comment.

Comment:

Dear authors,

Thank you for your submission. I think it was well written.

I have a small number of recommendations.

Method. Please include a reference for all test protocols - e.g. a citation for CMJ line 188 and do the same for all tests. Please add equations and calculations for:

- Boys PHV - Malian et al. 2020

- CMJ height - the Optojump uses the flight-time method outlined by Glatthorn 2011

- RSI mm/ms outlined by Flanagan and Comyns 2008

- Leg stiffness - equation can be found in the leg stiffness in female Soccer players by De Ste Croix and Lehnert

- Relative leg stiffness - McMahon and Cheng 1990

I think doing this would add to the rigour of the manuscript.

Response: We have added a formula for calculating the maturity offset to the manuscript. We agree that the description of the three tests involving some form of jump (CMJ, RSI, and leg stiffness) was somewhat vague and could have led to confusion. Therefore, we have added protocol specific references for each test. Additionally, we have included the formula needed to calculate RSI. In the case of CMJ and leg stiffness, we have omitted the formulas because describing the procedure requires a list of several formulas. Therefore, we prefer to refer readers to the literature. To review the changes in the methods section, please see the manuscript with tracked changes. We hope these changes have improved the rigor of the manuscript.

Comment: You should think about adding an abbreviation section - refer to Lloyd et al. 2009.

Response: Thank you for this suggestion. According to the PLOS Submission Guidelines, abbreviations should be defined upon first appearance in the text. A list of abbreviations is not required. Therefore, we have not included a separate section with abbreviations. However, we have carefully reviewed the manuscript to ensure that all abbreviations are defined upon their first appearance and are used consistently throughout the manuscript.

Comment: Statistical analysis. How was the sample size determined? G*Power? Detail the exact procedure. Please add this to Statical analysis section.

Response: Thank you for this comment. The following text has been added to the manuscript at the end of the section on statistical analysis:

“For logistical reasons, the sample size was limited by the availability of participants from a single soccer club. Therefore, no a priori power analysis was conducted. Instead, a posteriori sensitivity analysis was performed using the G*Power software version 3.1.9.7 (Heinrich-Heine-Universität, Düsseldorf, Germany). A two-tailed paired t-test was considered for the calculation, as the Fisher’s least significant difference test used in this study for pairwise comparisons is a special type of t-test. The following values were used for the calculation: significance level α = 0.05, power 1−β = 0.80, sample size n = 10 or 7. The required effect size was calculated to be d = 1.0 or 1.3, respectively.”

Comment: Discussion. I think you should include some the data from the r was ultra to back up discussion points.

Response: Thank you for this comment. Unfortunately, we were unable to determine the exact meaning of the term “r” in this comment. One possible interpretation is that this refers to the statistical software R, which, however, was not used in the manuscript. Another possible interpretation is that this refers to correlation coefficients between selected performance variables. However, correlation analysis was not part of the original study design, and in our opinion, its inclusion would not be consistent with the aim of the study. Furthermore, it would considerably expand the scope of an already extensive manuscript.

Comment: Also, I think you should cite some of the adaptations following PT - refer to Markovic and Mikulic 2010.

Response: Thank you for this suggestion. We understand this comment as a recommendation to better support in the discussion the mechanisms underlying the observed changes in performance following plyometric training. Therefore, we have added the review by Markovic and Mikulic (2010) to the relevant points in the discussion, which supports statements related to neuromuscular and musculoskeletal adaptations following plyometric training.

Comment: Limitations. A further limitation was that the study didn't include a mid-PHV. Can you detail what this might mean - would the results be similar in the mid-PHV group? If yes/no, why? Read the meta-analysis by Ramirez-Campillo and Sortwell 2023.

Response: Thank you for this important comment. We agree that the absence of a circa-PHV group is a limitation of the present study. Unfortunately, we did not have a sufficient number of participants in this maturity category to create a separate group for a valid statistical analysis. Therefore, we are unable to determine whether the response of circa-PHV players would be more similar to the pre- or post-PHV group. We have clarified this point in the limitations section as follows:

“Another limitation of the present study is the absence of a circa-PHV group, which limits the ability to assess how players around peak height velocity might have responded compared to pre-PHV and post-PHV groups.”

Reviewer #2

The study aimed to investigate whether the performance benefits of CMJ training are influenced by biological maturation status. This is a relevant and timely research question, particularly considering the importance of biological maturation in youth athletic development and training responsiveness.

However, although maturation status was included as a covariate, designing and implementing the training program according to the participants’ maturation stage might have allowed for a more precise evaluation of training adaptations and potentially revealed differential responses across maturation groups. Such an approach may also strengthen the practical implications for coaches and practitioners working with youth soccer players.

Overall, the manuscript is clearly written and methodologically sound. No further major comments or additional suggestions.

Response: We would like to thank the reviewer for the time he/she spent reviewing the manuscript and for their positive evaluation. Although the reviewer had no comments to incorporate, the manuscript was revised according to the comments of Reviewer 1, which we hope has further improved the quality and clarity of our manuscript.

---

## [Editor Report · Decision Letter 1]

23 Apr 2026

The effect of plyometric training on physical performance in youth soccer players: A randomized controlled trial with maturation status as a covariate

PONE-D-26-10103R1

Dear Dr. Krejčí,

We’re pleased to inform you that your manuscript has been judged scientifically suitable for publication and will be formally accepted for publication once it meets all outstanding technical requirements.

Kind regards,

Stefano Amatori, Ph.D.

Academic Editor

PLOS One
---

## [Editor Report · Acceptance letter]

PONE-D-26-10103R1

PLOS One

Dear Dr. Krejčí,

I'm pleased to inform you that your manuscript has been deemed suitable for publication in PLOS One. Congratulations! Your manuscript is now being handed over to our production team.

Kind regards,

on behalf of

Prof. Stefano Amatori

Academic Editor

PLOS One